# Functional Characterization of Endothelial Cells Differentiated from Porcine Epiblast Stem Cells

**DOI:** 10.3390/cells11091524

**Published:** 2022-05-02

**Authors:** Joon-Hong Shin, Bo-Gyeong Seo, In-Won Lee, Hyo-Jin Kim, Eun-Chan Seo, Kwang-Min Lee, Soo-Been Jeon, Sang-Ki Baek, Tae-Suk Kim, Jeong-Hyung Lee, Jung-Woo Choi, Cheol Hwangbo, Joon-Hee Lee

**Affiliations:** 1Department of Animal Bioscience, College of Agriculture and Life Sciences, Gyeongsang National University, Jinju 52828, Korea; leonardo6858@gnu.ac.kr (J.-H.S.); yi.innwon@gnu.ac.kr (I.-W.L.); sbub9598@naver.com (S.-B.J.); sangki.beak@kitox.re.kr (S.-K.B.); kts9347@gnu.ac.kr (T.-S.K.); 2Division of Applied Life Science (BK21), PMBBRC and Research Institute of Life Sciences, Gyeongsang National University, Jinju 52828, Korea; sbk6427@naver.com (B.-G.S.); jin4477@hanmail.net (H.-J.K.); eunchan3927@gmail.com (E.-C.S.); stare4567@naver.com (K.-M.L.); 3Division of Life Science, College of Natural Sciences, Gyeongsang National University, Jinju 52828, Korea; 4Department of Biochemistry, College of Natural Sciences, Kangwon National University, Chuncheon 24414, Korea; jhlee36@kangwon.ac.kr; 5College of Animal Life Science, Kangwon National University, Chuncheon 24414, Korea; jungwoo.kor@gmail.com; 6Institute of Agriculture & Life Science, College of Agriculture and Life Sciences, Gyeongsang National University, Jinju 52828, Korea

**Keywords:** porcine epiblast stem cells, endothelial cells, magnetic activated cell sorting, functional evaluation

## Abstract

Endothelial cells (ECs), lining blood vessels’ lumen, play an essential role in regulating vascular functions. As multifunctional components of vascular structures, pluripotent stem cells (PSCs) are the promising source for potential therapeutic applications in various vascular diseases. Our laboratory has previously established an approach for differentiating porcine epiblast stem cells (pEpiSCs) into ECs, representing an alternative and potentially superior cell source. However, the condition of pEpiSCs-derived ECs growth has yet to be determined, and whether pEpiSCs differentiate into functional ECs remained unclear. Changes in morphology, proliferation and functional endothelial marker were assessed in pEpiSCs-derived ECs in vitro. pEpiSCs-derived ECs were subjected to magnetic-activated cell sorting (MACS) to collect CD-31+ of ECs. We found that sorted ECs showed the highest proliferation rate in differentiation media in primary culture and M199 media in the subculture. Next, sorted ECs were examined for their ability to act as typical vascular ECs through capillary-like structure formation assay, Dil-acetylated low-density lipoprotein (Dil-Ac-LDL) uptake, and three-dimensional spheroid sprouting. Consequently, pEpiSCs-derived ECs function as typical vascular ECs, indicating that pEpiSC-derived ECs might be used to develop cell therapeutics for vascular disease.

## 1. Introduction

Endothelial cells (ECs), which constitute the lumen of blood vessels in the body, play a critical role in modulating vascular functions [1]. They are implicated in thrombosis and platelet adhesion, immunological and inflammatory responses, and vascular tone and blood flow regulation [2,3,4]. Endothelial dysfunction has been linked to a wide range of pathologic events, including atherosclerosis, congestive heart failure, and pulmonary hypertension [5,6,7]. Hence, ECs have been utilized in a variety of in vitro disease models to explore vascular dysfunction, such as diabetes and atherosclerosis development, coronary artery disease, and COVID-19 virus infection [8,9,10,11]. ECs play an essential role in vascular homeostasis by interacting with circulating cells and adjacent cells in the vessel walls. However, functional properties of vascular ECs are gradually reduced due to the loss of homeostasis caused by dietary preferences, smoking, aging and physical inactivity. ECs failure, which results in atherosclerosis, is the leading cause of cardiovascular disease (CVD) and its repercussions, such as heart attacks and strokes [12,13,14,15]. Therefore, vascular ECs regeneration avail of pluripotent stem cell (PSCs)-based therapy is an attractive therapeutic approach to the treatment of CVD because ECs essentially participate in any organ regeneration program [16,17].

PSCs retain the pluripotency to differentiate into any cell types in the body and the self-renewal capacity to replicate from mother cells to daughter cells indefinitely [18,19]. In general, PSCs include embryonic stem cells (ESCs) and epiblast stem cells (EpiSCs) derived from particular cell mass (ICM or epiblast) of pre- or post-implantation embryos [20,21,22]. Furthermore, induced pluripotent stem cells (iPSCs) reprogrammed from terminally differentiated somatic cells by transduction of exogenous reprogramming factors [23,24,25]. In the mouse, ESCs and EpiSCs established from pre- or post-implantation embryos represented different cytokine dependency to maintain the pluripotent states termed as “naïve” and “primed” [20,26,27,28,29,30]. Depending on the species, it was also that murine ESCs and human ESCs demand different signaling pathways to maintain their pluripotent states [27,30,31,32]. Although PSCs have significantly different characteristics from sources and species, porcine epiblast stem cells (pEpiSCs) represented a feature of the primed state like murine EpiSCs and human ESCs [33,34].

Since then, ECs were first successfully generated from human ESCs [35], a variety of differentiation protocols have been extensively studied to generate ECs from human PSCs mainly applying sequential addition in culture of growth factors such as bone morphogenetic protein-4 (BMP-4) and vascular endothelial growth factor (VEGF) for the development of vascular disease therapy [36,37]. However, there are highly harsh reactions, such as ethical and stability concerns to apply human ESCs for human disease. As an alternative to theses, we recently reported that an in vitro differentiation of pEpiSCs into ECs is efficiently established for applying the treatment of vascular diseases in humans [38]. The cells differentiated from pEpiSCs, on the other hand, were highly heterogeneous mixed with undifferentiated PSCs and differentiated ECs. It may potentially cause teratoma formation following transplantation into the body, which would exclude direct clinical applications of PSCs. Therefore, there has been a strong emphasis on the purification of contractile ECs from ESCs, such as using Percoll gradient fractionation, fluorescence-activated cell sorting (FACS) separation of CD-31+ cells, surface expression of signal-regulatory protein and activated leukocyte cell adhesion molecule [39,40,41,42].

Addressing this current limitation, we describe an efficient procedure for selecting ECs solely from cell mixtures differentiated from pEpiSCs using magnetic beads labeled with an endothelial cell marker CD-31, providing growth conditions for these cells to proliferate effectively. The sorted ECs show typical features of ECs, such as capillary-like structure formation, Dil-labelled acetylated low-density lipoprotein (Dil-Ac-LDL) uptake and sprouting ability in three-dimensional spheroid. Moreover, the cells maintain their functional properties during prolonged culture through at least ten passages.

## 2. Materials and Methods

### 2.1. Cells and Cell Culture

Culture of porcine epiblast stem cells (pEpiSCs) were used for this experiment as described in a previous study [34]. Briefly, pEpiSCs were derived from epiblasts that dissociated from in vivo embryos collected nine days after insemination. Mouse embryonic fibroblast cells (MEFs) were derived from fetal mice at 11.0 days in placentas of pregnant female mouse. MEFs were cultured in Dulbecco’s modified eagle’s medium (DMEM/F12) supplemented with 10% of fetal bovine serum (FBS), 1.5 mM of β-mercaptoethanol, 1% of MEM and 1% of penicillin/streptomycin. When MEFs showed about 80% confluence, MEFs were inactivated by 10 µg/mL mitomycin C for 2½ h. Inactivated MEFs (iMEFs) were treated with 0.05% Trypsin/EDTA, which was used to detach the cells, 1.2 × 10^6^ cells seeded on plate. After 24 h, pEpiSCs cultured on iMEFs with DMEM/F-12 supplemented with 20% of FBS, 1% of GlutaMAX, 1% of MEM, 1% of penicillin/streptomycin and 0.1 mM β-mercaptoethanol and incubated at 39 °C in 5% CO_2_. pEpiSCs were passaged using mixture of 1 mg/mL Collagenase IV and 1 mg/mL Dispase for 10 min at 39 °C.

### 2.2. Reagents and Antibodies

Dulbecco’s modified eagle’s medium (DMEM/F12; 11995-073), 0.05% Trypsin/EDTA (25300-54), Collagenase IV (17104-019), Dispase (17105-041), M199 (M4530) were purchased from GIBCO (Grand Island, NY, USA). Fetal bovine serum (FBS; S001-01) used for MEF culture was purchased from WELGENE (Gyeongsan, Korea). β-mercaptoethanol (M3148), MEM (M7145), penicillin/streptomycin (P0781), mitomycin C (M4287), bovine serum albumin (BSA, A6003), heparin (H3149), paraformaldehyde (PFA; P6148), methyl cellulose (M7140), HEPES (391340) were purchased from Sigma (St. Louis, MO, USA). Endothelial cell growth basal medium-2 (EBM-2; CC-3156), endothelial cell growth medium-2 SingleQuots^®^ (EGM-2; CC4176) were purchased from Lonza (Basel, Switzerland). Vascular endothelial cell growth factors (VEGF; 293-VE) was purchased from R&D (Minneapolis, MN, USA). Matrigel^®^ growth factor reduced (354230) used for differentiation, matrigel^®^ (354234) used for capillary-like structure formation assay, endothelial cell growth supplement (ECGS; 356006), collagen I (354249) were purchased from Corning (Corning, NY, USA). Dynabeads™ M-280 streptavidin Sheep anti-rabbit (11203D) was purchased from Thermo Fisher Scientific (Grand Island, NY, USA). CD31 antibody (NB100-2284), Ki-67 antibody (NB500-170) were purchased Novusbio (Centennial, CO, USA). Ethylene-diamine-tetraacetic acid (EDTA; EDT001.500) was purchased from BioShop (Ontario, Canada). FBS (TMS-013-BKR) used for solely EC culture was purchased from Merck (Darmstadt, Germany). Phycoerythrin (PE) conjugated CD-31 antibodies (555027) was purchased from BD Pharmingen (Becton Dickinson, NJ, USA). Alexa Fluor^®^ 546 Goat Anti-Rabbit IgG (A11010), Phalloidin (A12379) were purchased from Invitrogen (Carlsbad, CA, USA). Hoechst 33342 was purchased from Life Technologies (Prederick, MD, USA). RNeasy Plus Mini Kit (74134) was purchased from QIAGEN (Valencia, CA, USA). Revoscript™ RT Premix (25087) was purchased from iNtRON Biotechnology Inc (Seongnam, Korea). GoTaq^®^ SYBR Master Mix (QPK201) was purchased from Promega (Madison, WI, USA). NaOH (39155-1250) was purchased from Junsei Chemical (Tokyo, Japan). Dil-Ac-LDL (022K) was purchased from Cell applications Inc. (H-1000; San Diego, CA, USA) was purchased from Vectorlabs (Burlingame, CA, USA).

### 2.3. In Vitro Differentiation of Endothelial Cells from Porcine Epiblast Stem Cells

pEpiSCs were cultured on iMEFs for 3 days, then detached stem cell colonies using manual picking. The pEpiSCs were differentiated into endothelial cells (ECs) in 50 ng/mL of VEGF included EGM-2 with VEGF excluded endothelial cell growth medium-2 SingleQuots^®^. Differentiation has proceeded for 8 days on culture plates coated with matrigel^®^ (1:40 dilution with DMEM/F-12 medium) at 39 °C.

### 2.4. Magnetic-Activated Cells Soring (MACS)

Dynabeads™ M-280 streptavidin Sheep anti-rabbit and CD31 antibody were mixed in 1:10 ratio and incubated at 4 °C for overnight. ECs differentiated from pEpiSCs were washed with Dulbecco′s phosphate-buffered saline (D-PBS). The cells were detached in 2 mM of ethylene-diamine-tetraacetic acid (EDTA) for 15 min at 39 °C and then centrifuged at 300× *g* for 3 min. Collected cells were suspended in 1% bovine serum albumin-DMEM medium. Magnetic beads labeled with CD-31 antibody were washed three times with 1% BSA/DMEM using PolyATtract^®^ system 1000 magnetic separation stand (Promega, Z5410). Suspended cells in 1% BSA/DMEM and magnetic beads labeled with the antibody were mixed and then rotated at room temperature for 1 h. Then, the mixtures were washed with 1% BSA/DMEM to remove unlabeled cells using a magnetic stand five times. Sorted ECs were seeded on culture plates coated with 0.2% gelatin for the primary culture. EGM-2, M199 supplemented with 20% of FBS, 30 μg/mL of endothelial cell growth supplement (ECGS) and 100 μg/mL of heparin were used for the culture medium of sorted ECs.

### 2.5. Proliferation Assay

Cells were seeded on culture plates coated with 0.2% gelatin and then counted the cell numbers for 5 days. Additionally, cell proliferation rates were examined with the expression of Ki-67 as a representative proliferation nuclear marker [43].

### 2.6. Flow Cytometry Analysis

Cells were washed with DPBS and then treated with 2 mM EDTA/PBS at 37 °C for 10 min. After centrifugation at 300× *g* for 3 min, collected purified ECs were suspended in stain buffer consisting of 2% BSA in PBS. The cells were stained with phycoerythrin (PE) conjugated CD-31 antibodies for 1 h at room temperature in the dark. The cells were suspended in stain buffer and then analyzed by FACSverse™ (BD Biosciences).

### 2.7. Immunocytochemistry

Cells were fixed with 4% paraformaldehyde (PFA) for 20 min. The cells were treated with the blocking solution (5% BSA/PBS-T) for 1 h, and then incubated with CD-31 antibody at 4 °C for 16 h, followed by Alexa Fluor^®^ 546 Goat Anti-Rabbit IgG antibody. Hoechst 33342 was used to nucleus counterstain. All images were acquired using the LEICA fluorescence microscope (LEICA, DM 2500) and performed with the Leica Application Suite (LAS; LEICA, version 3.8). 

### 2.8. Quantitative Polymerase Chain Reaction

Total RNA was extracted using RNeasy Plus Mini Kit following the manufacturer’s methods. Extracted RNAs were synthesized into cDNA using the Revoscript™ RT Premix. Quantitative real-time polymerase chain reaction (q-PCR) was performed using the GoTaq^®^ SYBR Master Mix with Rotor-Gene Q-Pure Detection system (QIAGEN). The primer list used for quantitative real-time polymerase chain reaction is Table 1. The gene expression was quantified relative to the reference gene (18S).

### 2.9. Three-Dimensional Spheroid Sprouting Assay

Three-dimension spheroid sprouting of purified ECs was performed as described in the previous study [44]. Cells were separated into single cells with 0.05% trypsin/EDTA. Spheroids were formed using methocel solutions consisting of 3 g of methyl cellulose in 125 mL of M199. Single cells were counted to 500 cells per 1 spheroid in 25 µL droplet with 20% methocel solutions in each medium. Droplets were formed on the inverted lid of 100 mm culture dishes and then incubated at 37 °C for 24 h. Droplets formation of spheroids were collected from dish lids with PBS containing 10% FBS. Collected spheroids were centrifuged at 100× *g* for 5 min. For embedding in collagen of spheroids, collagen solution was mixed with acetic acid, 100 mg/mL of collagen I and M199 in a 4:4:1 ratio. Collagen solutions and 80% of methocel were mixed in a 1:1 ratio and then added to spheroids. Mixtures were deposited to 24 well culture plates and then polymerized at 37 °C for 1 h. When mixtures were polymerized, 330 ug/mL ECGS in M199 medium was added to mixtures to induce ECs sprouting. After 24 h, the spheroids in polymerized collagen mixtures were fixed in 4% PFA for 20 min. Phalloidin was stained in mixtures (1:250, Invitrogen, A12379). All images were acquired using OPTIKA fluorescence microscope (OPTIKA, XDS-3FL4) and performed with the software (OPTIKA, vision pro). Sprouts length was calculated using the ImageJ software.

### 2.10. Capillary-Like Structure Formation Assay

To capillary-like structure formation, cells were cultured on Matrigel, thawed at 4 °C for overnight and 50 μL added to Matrigel on 96 wells plate. Plates coated with Matrigel were incubated at 37 °C for 30 min. Cells were counted to 2 × 10^4^ and then seeded on Matrigel. All images were acquired using an Olympus fluorescence microscope (Olympus, DP70) and performed with the DP manager (Olympus, version 3.1.1.208).

### 2.11. Dil-Acetylated-LDL Uptake Assay

Cells were added 2 μg/mL of Dil-Ac-LDL and then incubated at 37 °C for 4½ h. The cells were fixed with 4% PFA for 10 min and then washed with PBS. Hoechst 33342 was used for nuclear staining. All images were explored using the LEICA fluorescence microscope (LEICA, DM 2500) and performed with the Leica Application Suite (LAS; LEICA, version 3.8).

### 2.12. Statistical Analysis

Graph Pad Prism software v7.00 (GraphPad) was used to analysis of data. Relative mRNA levels of OCT-3/4, NANOG, SOX2, quantification of Ki-67 positive cells in culture of differentiation media, sprouts length, branch points and quantification of Dil-Ac LDL uptake assay were analyzed in triplicate and data were presented as means ± SEM. One-way or two-way ANOVA used statistical significance between groups. *P* < 0.05 was considered statistically significant.

## 3. Results

### 3.1. Separating Endothelial Cells Differentiated from pEpiSCs

Endothelial cells (ECs) were differentiated from porcine epiblast stem cells (pEpiSCs) using the method described in our previous study [38]. We sought to separate ECs solely from differentiated pEpiSCs by using magnetic-activated cell sorting (MACS) with CD-31 antibody, an endothelial cell marker (Figure 1A), and the expression of CD-31 was analyzed by flow cytometry and immunofluorescence. CD-31 expression was found in about 28% of the unsorted cell population, whereas 100% in the sorted cell population (Figure 1B). Furthermore, unsorted cells showed partial CD-31 expression, but sorted ECs showed most CD-31 expression by immunofluorescence (Figure 1C). pEpiSCs did not express at all. To evaluate pluripotency of sorted ECs, gene expression of pluripotency markers such as OCT-3/4, NANOG and SOX2 were measured. Comparison of the pluripotency in pEpiSCs, unsorted ECs and sorted ECs revealed significant changes. These changes showed that pluripotency markers were significantly decreased in sorted ECs compared to pEpiSCs (Figure 1D). These results indicate that MACS-based cell sorting is sufficient for separating ECs solely and that sorted ECs lost their pluripotency.

### 3.2. Proliferation of pEpiSCs-Derived ECs

For the primary culture of sorted ECs, these ECs were cultured using various media for a couple of days (Table 2). Sorted ECs were not primarily survived and proliferated in EGM-2 or M199 while growing well in differentiation media (Figure 2A). Photographs were obtained on the day using a phase-contrast microscope, with representative photographs shown. Interestingly, after subculture of these cells, sorted ECs were grown well in M199. The proliferation rate of subculture of ECs were evaluated for five days. The cell growth in each culture condition was compared to differentiation media as the control. As the results, EGM-2 and EGM-2-EV culture showed lower proliferation than control otherwise, M199 showed a significantly increased proliferation rate than control on days four and five. Also, the cells were stained with Ki-67 and showed the highest staining level when cultured in M199 compared to other media (Figure 2B). These results suggested that early passage of sorted ECs are needed differentiation media for stabilization and then, M199 is the best condition for growth and proliferation.

### 3.3. Angiogenic Function of pEpiSCs-Derived ECs by Three-Dimensional Spheroid Sprouting

Typical ECs have a function of angiogenesis, which is to form new vessels by various signaling. To confirm the angiogenesis capacity of differentiated cells in three-dimensional conditions, spheroids spouting assay derived from three types of cells (pEpiSCs, unsorted ECs and sorted ECs) (Figure 3). pEpiSCs and sorted ECs formed well spheroids, but unsorted ECs did not form spheroids. After allowing the spheroids to sprout, capillary-like structures sprouted out of spheroids were formed in sorted ECs (Figure 3A). However, no branched spheroids were displayed from pEpiSCs and unsorted ECs coexisting with undifferentiated PSCs and differentiated ECs. Additionally, sprouting out of three-dimensional spheroids derived from the sorted ECs were identified with fluorescence staining (Figure 3B,C). Collectively, these results suggested that pEpiSCs-derived ECs have an angiogenetic function as mature vascular endothelial cells in three-dimensional conditions.

### 3.4. Vessel Organization of pEpiSCs-Derived ECs Using Capillary-like Structure Formation

Capillary-like structure formation was performed to evaluate angiogenesis’s reorganization in vascular ECs. This assay was conducted to identify the functional capability of sorted ECs as acts typical ECs. The ability to form a capillary-like structure was assessed by seeding pEpiSCs, differentiated endothelial cells on matrigel-coated plates (Figure 4). As a result, sorted ECs started to form capillary-like structures for two hours. Capillary-like structures from sorted ECs were gradually expanded and then widely broadened. By contrast, unsorted ECs presented unclear capillary-like structure formation because differentiated ECs and undifferentiated pEpiSCs were mixed (Figure 4A,B). Interestingly, single cells derived from pEpiSCs colonies formed partial capillary-like structures. These results show that pEpiSCs-derived ECs have reorganization of vessel tube capability as functional vascular endothelial cells.

### 3.5. Acetylated Low Density Lipoprotein Uptake of pEpiSCs-Derived ECs

ECs maintain homeostasis of cholesterol concentration in the blood vessels by uptaking acetylated low-density lipoprotein (Ac-LDL) [45]. To evaluate the function of Ac-LDL uptake in pEpiSCs, unsorted ECs or sorted ECs were examined with Dil-Ac-LDL (Figure 5). As a result, Dil-Ac-LDL uptake was detected only in differentiated endothelial cells (unsorted ECs and sorted ECs). Unsorted ECs showed little uptake of Dil-Ac-LDL, especially sorted ECs were shown the highest uptake of Dil-Ac-LDL. These results revealed that differentiated endothelial cells have a capacity of Ac-LDL uptake as functional property of endothelial cells.

## 4. Discussion

Technologies for the production of a vessel-, tissue-, organ-, disease-, and further patient-specific ECs will become a fundamental necessity for research of molecular target validation, high-throughput drug screening and ECs-based cell therapy, including the engineering of clinically applicable engineered vascular tissue grafts. Although primary ECs have several limitations, such as restricted scalability and high probability of karyotypic defects, ECs have been used in various disease models to explore vascular dysfunction. Because of their indefinite self-renewal and high pluripotency, pluripotent stem cells (PSCs), which include embryonic stem cells (ESCs), epiblast stem cells (EpiSCs), and induced pluripotent stem cells (iPSCs), are the promising therapeutic strategy for human degenerative diseases. However, owing to the harsh ethical and limitation of culture expansion to utilize human ESCs (hECS) for regenerative medicine, many attention has been diverted to non-primate species such as pig retained anatomical and physiological similarities with humans [46]. Accordingly, porcine PSCs were of considerable significance for human degenerative diseases therapy [47].

We recently reported an in vitro differentiation protocol in which pEpiSCs were incubated stem cell culture medium on a feeder layer of mitomycin-treated mouse embryo fibroblasts (MEFs) for three days. Then, the condition changed to EBM-2-EV supplemented with 50 ng/mL VEGF on matrigel for eight days to induce ECs differentiation [38]. We observed differentiation efficiencies of approximately 27% for CD31-positive cells. Such protocols are undoubtedly scalable; however, strategies to select pure ECs with culture conditions that allow pure ECs to proliferate effectively and further functional validation of pEpiSC-derived ECs into typical vascular ECs would be essential for therapeutic application in a clinical trial. Applying the separation of ECs from the heterogeneous mixtures, pEpiSCs-derived cells, including differentiated or undifferentiated ECs were adopted in magnetic-activated cell sorting (MACS) using CD31 antibody. Although the protocol achieved the approach yield of about 27% ECs, 100% purified ECs could be identified through flow cytometry analysis by MACS sorting These were achieved by generating a more efficient amount that is not limited by the current scalable culture conditions of primary ECs.

For the widespread application of ECs, it has been fundamentally important to establish an efficient ECs proliferation system. However, sorted ECs were not proliferated effectively in Endothelial Cell Growth Medium-2 BulletKit (EGM-2) as a specialized ECs growth media. It was reported that adding 50 ng/mL of VEGF to EGM-2-EV supports the ECs survival for the primary culture until the primary sorted cells start growing at the normal proliferation rate [48]. When the required growth factors were not added to the culture medium, proliferation of endothelial progenitor cells (EPCs) proceeded slowly and cell death occurred lastly [49]. For such reasons, it was necessary to culture the sorted ECs in differentiation media until the primary passage. In general, VEGF plays an essential role in the survival and proliferation of ECs by activating of PI3K/Akt/forkhead signaling pathway in scalable suspension culture [50,51,52]. After the primary culture in differentiation media, the M199 culture system-induced most significant effect on the proliferation of ECs. Therefore, it is reasonable that the requirement of cytokines is different during the ECs proliferation. Collectively, the purified ECs were identified to exhibit the greatest effect on VEGF for the primary culture and the M199 culture system for the proliferation. We further confirmed the expression of Ki67, a marker of cell proliferation, through fluorescence staining.

Considering the requirement of practical grade for the vascular disease therapy, functional assessment of generated pEpiSCs-derived ECs were evaluated applying three different assays. ECs enriched by surface marker selection would provide a safer cell resource. Expression of CD-31 as an ECs-specific surface marker was observed strongly in early vascular development and capillary-like structures derived from ECs on Matrigel [53,54]. Although a previous report described ECs enrichment from human ESCs with a selection of CD-31+ expression [55], the purity of enriched ECs in that study was lower (~20%) than ours (~30%) [38]. On the other hand, PSCs differentiation usually occurs within multicellular, three-dimensional structures called embryoid bodies (EBs). However, in the current study, ECs sorted by selecting CD-31+ expression assembled networks of capillary-like structures, whereas pEpiSCs and unsorted ECs differentiated from pEpiSCs were barely formed. As well as, pEpiSCs-derived ECs solely networks showed much branching points compared with networks from pEpiSCs or unsorted differentiated ECs. The sorted ECs on Matrigel were attached and wrapped around in a way that is reminiscent of angiogenesis.

Additionally, to characterize the phenotypic nature of the ECs derived from pEpiSCs, a functional method that involves measuring Ac-LDL uptake using the fluorescent probe Dil (Dil-Ac-LDL) was performed. The sorted ECs were brilliantly fluorescent, whereas the fluorescent intensity of pEpiSCs and unsorted ECs were barely detectable. Finally, the three-dimensional spheroid sprouting of sorted ECs using the hanging drop protocol cultured in collagen type IV mixtures supplemented with ECGS for one day was examined. Nascent capillary-like structures out of the three-dimensional spheroid were formed in the sorted ECs, elucidating the vessel formation. These suggests that the sorted ECs by selected CD-31+ expression were fully differentiated and functionally competent.

## 5. Conclusions

The protocol described here offers the first opportunity to generate purified ECs from pEpiSCs with well-set up culture conditions for proliferation, which show the functionality of typical vascular ECs. Functional tests revealed that the generated ECs might be used in vitro assays to examine angiogenesis or cellular responses to various vascular diseases. Additionally, the ability to generate functional-ECs in sufficient quantities for cell therapy techniques may allow these purified pEpiSCs-derived ECs to be used in regenerative treatments.

## Figures and Tables

**Figure 1 cells-11-01524-f001:**
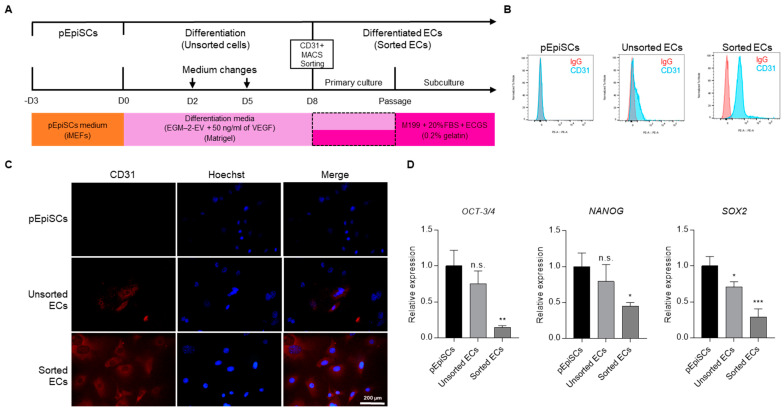
Establishment of differentiated ECs sorting. (**A**) The schematic diagram presented in vitro differentiation of pEpiSCs into ECs in differentiation media on matrigel for eight days. On day 8, ECs were separated from cell mixtures differentiated from pEpiSCs using magnetic beads coated with endothelial cell marker (CD-31). After the cell sorting, ECs were cultured in differentiation media for the primary culture and then proliferated in M199 on 0.2% Gelatin. (**B**) Flow cytometry analysis of CD-31 expression in pEpiSCs, unsorted ECs and sorted ECs. (**C**) Immunofluorescence of CD-31 in pEpiSCs, unsorted ECs and sorted ECs. Blue: staining of Hoechst, Red: staining of CD-31. Scale bar = 200 μm. (**D**) Relative mRNA levels of OCT-3/4, NANOG and SOX2 in pEpiSCs, unsorted ECs and sorted ECs. Values presented as mean SEM. * *p* < 0.05, ** *p* < 0.01, *** *p* < 0.001 vs. pEpiSCs, n.s.: not significant.

**Figure 2 cells-11-01524-f002:**
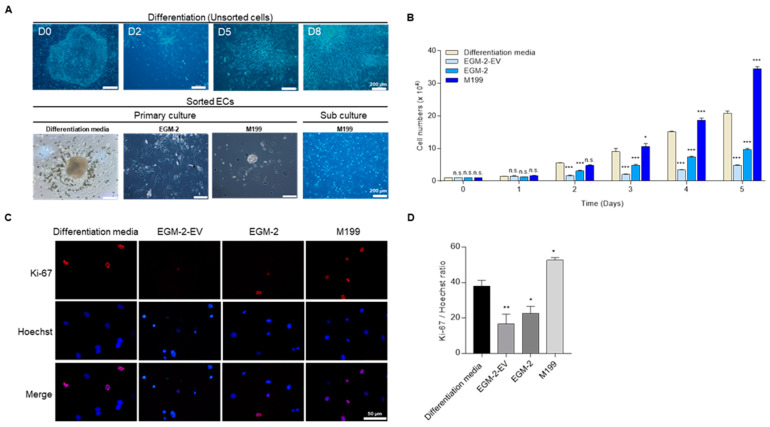
The proliferation of sorted ECs. (**A**) Morphologies of pEpiSCs and differentiated cells in differentiation media on the days 0, 2, 5, 8, respectively. Sorted ECs were cultured in differentiation media, EGM-2, or M199 culture system on 0.2% gelatin for the primary culture. Primary sorted ECs cultured in differentiation media were transferred to M199 culture system for sub-culture. (**B**) Proliferation rates of 1.0 × 10^4^ of sorted ECs were evaluated in four culture conditions. Values presented as mean SEM. * *p* < 0.05, ** *p* < 0.01, *** *p* < 0.001 vs. differentiation media, n.s.: not significant. (**C**) Immunofluorescence of Ki-67 in differentiation media, EBM-2-EV, EGM-2 and M199. Red: staining of Ki-67, blue: staining of Hoechst. Scale bar = 50 μm (**D**) Quantification of Ki-67 positive cells in culture of sorted ECs in differentiation media, EBM-2-EV, EGM-2 and M199. Values presented as mean SEM. * *p* < 0.05, ** *p* < 0.01 vs. differentiation media, n.s.: not significant.

**Figure 3 cells-11-01524-f003:**
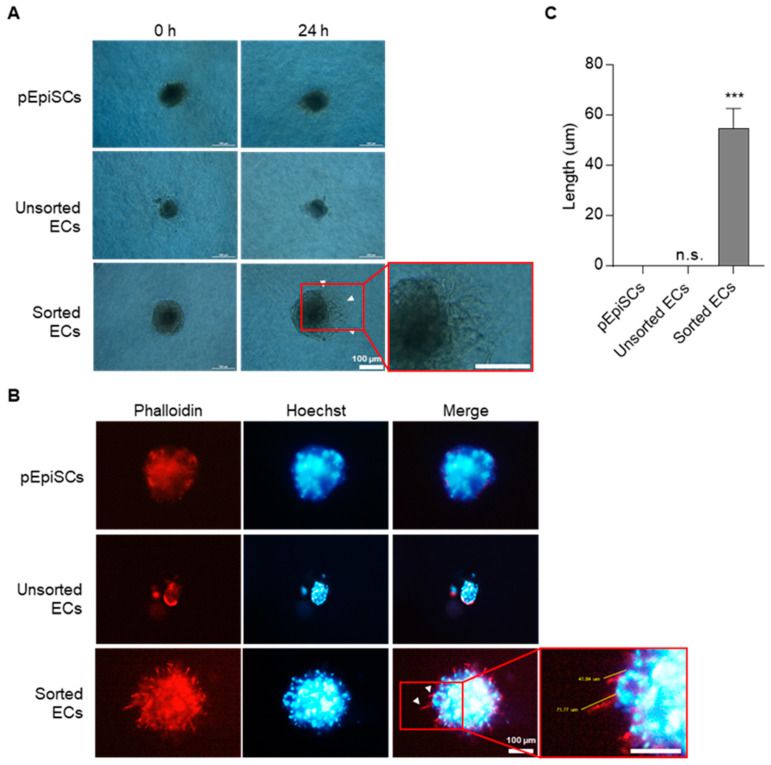
Three-dimensional spheroid sprouting of sorted ECs. (**A**) Three-dimensional spheroids derived from pEpiSCs, unsorted ECs and sorted ECs using the hanging drop protocol were cultured in collagen type I mixtures supplemented with ECGS for 24 h. Arrowheads indicated sprouting (capillary-like) structure out of three-dimensional spheroid derived from the sorted ECs. Scale bar = 100 μm. (**B**) Phalloidin staining in three-dimensional spheroid sprouting. Arrowheads indicated a capillary-like structure (sprouting) out of three-dimensional spheroid derived from the sorted ECs. Blue: staining of Hoechst, Red: staining of phalloidin. Scale bar = 200 μm. (**C**) Sprouts length was calculated by using ImageJ. Values presented as mean SEM. *** *p* < 0.001 vs. pEpiSCs, n.s.: not significant.

**Figure 4 cells-11-01524-f004:**
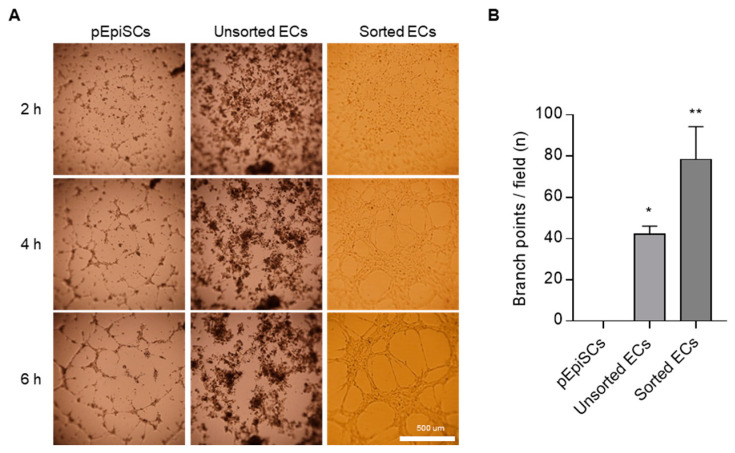
Formation of capillary-like structure of sorted ECs. (**A**) 2.0 × 10^4^ of cells were cultured in 96-well cell culture plates on matrigel. Formation of capillary-like structures of sorted ECs was observed on matrigel for 6 h. Scale bar = 500 μm. (**B**) Branch points of pEpiSCs, unsorted ECs and sorted EC was counted. Values presented as mean SEM. * *p* < 0.05, ** *p* < 0.01 vs. pEpiSCs, n.s.: not significant.

**Figure 5 cells-11-01524-f005:**
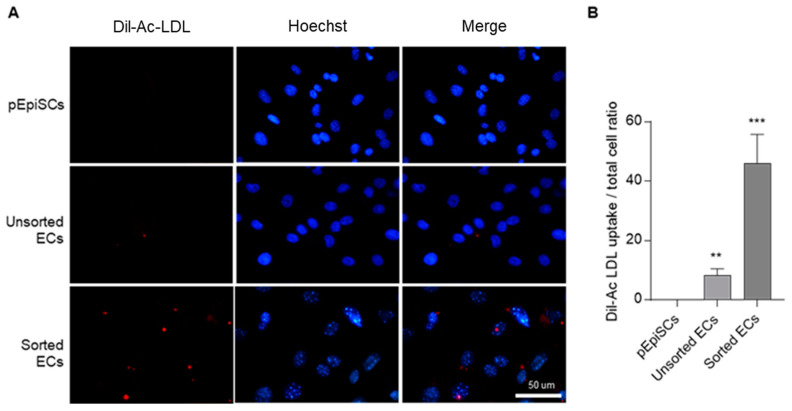
Uptake of Dil-acetylated low-density lipoprotein of Sorted ECs. (**A**) Dil-Ac-LDL uptakes of pEpiSCs, unsorted ECs and sorted ECs were examined at 4.5 h. Blue: staining of Hoechst. Red: staining of Dil-Ac-LDL. Scale bar = 50 μm. (**B**) Quantification of Dil-Ac LDL uptake assay in pEpiSCs, unsorted ECs and sorted EC. Values presented as mean SEM. ** *p* < 0.01, *** *p* < 0.001 vs. pEpiSCs, n.s.: not significant.

**Table 1 cells-11-01524-t001:** Sequences of primers used in q-RT-PCR.

Gene	Sequence (5′→3′)		References
	Forward	Reverse	
18S	TCG GAA CTG AGG CCA TGA TT	GAA TTT CAC CTC TAG CGG CG	NR_046241.1
OCT-3/4	GGA TAT ACC CAG GCC GAT GT	GTC GTT TGG CTG AAC ACC TT	NM_001113060.1
NANOG	CCC GAA GCA TCC ATT TCC AG	GAT GAC ATC TGC AAG GAG GC	DQ_447201.1
SOX2	CAT GTC CCA GCA CTA CCA GA	GAG AGA GGC AGT GTA CCG TT	NM_001123197.1

**Table 2 cells-11-01524-t002:** Culture conditions for endothelial cells differentiated from porcine epiblast stem cells.

Name	Media	Supplements & Growth Factors
EGM-2	EBM-2	EGM-2 SingleQuot Kit
EGM-2-EV	EBM-2	VEGF excluded EGM-2 SingleQuot Kit
Differentiation Media	EBM-2	50 ng/mL VEGF-165, VEGF excluded EGM-2 SingleQuot Kit
M199	M199	20% PBS, 100 µg/mL heparin, 30 µg/mL ECGS

## Data Availability

Data are contained within the article.

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
