# Peer review of "Functional Characterization of Endothelial Cells Differentiated from Porcine Epiblast Stem Cells"

_cells, 2022, doi:10.3390/cells11091524_

Round 1
Reviewer 1 Report
Shin et la present an original research article entitled "Functional Characterization of Endothelial Cells Differentiated from Porcine Epiblast Stem Cells".
They look at the conditions to grow pEpiSCs-derived ECs and to differentiate them into functional ECs in order to develop these cells for therapeutics for vascular disease. They found that found that sorted ECs showed the highest proliferation rate in differentiation media in primary culture and M199 media in the subculture. They also showed their ability to act as typical vascular ECs through capillary- like structure formation assay, Dil-acetylated low-density lipoprotein uptake, and three-dimensional spheroid sprouting.
The paper is badly written and should be utterly corrected.
More endothelial sepcific markers such as syndecan-1, MMP7, VCMA1 should be dosed.
an in vivo model should be performed to establish proof of concept that these cells are able to establish themselves in the model.
Minor
abstract : "ECs), lining blood vessels’ lumen, play an essential ROLE in regulating vascular functions."
line 125 correct "Primary sorted ECs were not survived "
line 147. Angiogentic? or angiogenic
line 212 rephrase please "However, due to the harsh ethical, stability reactions and limitation of culture expansion to utilize human ESCs (hECS) for regenerative medicine,"
Author Response
Shin et la present an original research article entitled "Functional Characterization of Endothelial Cells Differentiated from Porcine Epiblast Stem Cells".
They look at the conditions to grow pEpiSCs-derived ECs and to differentiate them into functional ECs in order to develop these cells for therapeutics for vascular disease. They found that found that sorted ECs showed the highest proliferation rate in differentiation media in primary culture and M199 media in the subculture. They also showed their ability to act as typical vascular ECs through capillary- like structure formation assay, Dil-acetylated low-density lipoprotein uptake, and three-dimensional spheroid sprouting.
Thank you for reviewing this manuscript "Functional Characterization of Endothelial Cells Differentiated from Porcine Epiblast Stem Cells" which submitted to Cells. The manuscript has been revised according to the reviewer’s comments.
The paper is badly written and should be utterly corrected.
This manuscript has been proof read by a professional English editing service for revision.
More endothelial specific markers such as syndecan-1, MMP7, VCMA1 should be dosed.
CD31, like Oct-3/4 for pluripotency, is the most important marker for ECs. I don’t think to examine another markers such as syndecan-1, MMP7 and VCMA1 for ECs.
An in vivo model should be performed to establish proof of concept that these cells are able to establish themselves in the model.
In vivo functional characterization of endothelial Cells differentiated from porcine epiblast stem cells has been performed using immunodeficiency mice. So far, many results from this in vivo experiment are quite similar with in vitro results.
Minor
abstract : "ECs), lining blood vessels’ lumen, play an essential ROLE in regulating vascular functions."
We have put “role” in the sentence, as you mentioned.
line 125 correct "Primary sorted ECs were not survived "
We have altered Primary sorted ECs were not survived in line 125 into Sorted ECs were not primarily survived, as you mentioned.
line 147. Angiogentic? or angiogenic
Angiogenetic has been altered into angiogenic, as you mentioned.
line 212 rephrase please "However, due to the harsh ethical, stability reactions and limitation of culture expansion to utilize human ESCs (hECS) for regenerative medicine,"
The sentence ‘However, due to the harsh ethical, stability reactions and limitation of culture expansion to utilize human ESCs (hECS) for regenerative medicine’ has been altered into ‘However, owing to the harsh ethical and limitation of culture expansion to utilize human ESCs (hECS) for regenerative medicine’, as you mentioned.
Reviewer 2 Report
In this original research manuscript, Shin et al. describe the purification and functional characterization of porcine endothelial cells (ECs) generated by differentiation of porcine epiblast stem cells. The authors developed a protocol for magnetic cell sorting of CD31+ ECs and showed that it allowed for obtaining of highly enriched population. Further, the ECs could be expanded by optimization of the culture medium using M199. Finally, the cells showed ability for maintaining their functional capacities after prolonged culture as assessed by acetylated low-density lipoprotein uptake, three-dimensional spheroid sprouting, and capillary-like structure formation assays.
The manuscript is clearly written and easy to follow, although some sentences would benefit from style editing, e.g. short sentences such as “pEpiSCs did not express at all.” (line 107), or “Comparison of the pluripotency in pEpiSCs, unsorted ECs and sorted ECs revealed significant changes.” (lines 109-110) – it would be better to write full sentences indicating what the cells did not express, or what the significant changes were in.
The sentence „ However, the condition of pEpiSCs-derived ECs 31 growth has yet to be determined, and whether pEpiSCs differentiate into functional ECs as typical ECs remained unclear.” is repeated twice in the Abstract.
Since the highlight of the manuscript is focused on the relatively easier magnetic bead sorting, which was earlier done by FACS, and the following verification process has been well established, the results are sufficient to support the manuscript conclusions. Therefore, pending the moderate writing style editing, I would recommend the manuscript to be accepted.
Author Response
In this original research manuscript, Shin et al. describe the purification and functional characterization of porcine endothelial cells (ECs) generated by differentiation of porcine epiblast stem cells. The authors developed a protocol for magnetic cell sorting of CD31+ ECs and showed that it allowed for obtaining of highly enriched population. Further, the ECs could be expanded by optimization of the culture medium using M199. Finally, the cells showed ability for maintaining their functional capacities after prolonged culture as assessed by acetylated low-density lipoprotein uptake, three-dimensional spheroid sprouting, and capillary-like structure formation assays.
The manuscript is clearly written and easy to follow, although some sentences would benefit from style editing, e.g. short sentences such as “pEpiSCs did not express at all.” (line 107), or “Comparison of the pluripotency in pEpiSCs, unsorted ECs and sorted ECs revealed significant changes.” (lines 109-110) – it would be better to write full sentences indicating what the cells did not express, or what the significant changes were in.
Thank you for reviewing this manuscript "Functional Characterization of Endothelial Cells Differentiated from Porcine Epiblast Stem Cells" which submitted to Cells. The manuscript has been revised according to the reviewer’s comments.
The sentence „ However, the condition of pEpiSCs-derived ECs 31 growth has yet to be determined, and whether pEpiSCs differentiate into functional ECs as typical ECs remained unclear.” is repeated twice in the Abstract.
We have removed the repeated twice sentence in the abstract, as you mentioned.
Since the highlight of the manuscript is focused on the relatively easier magnetic bead sorting, which was earlier done by FACS, and the following verification process has been well established, the results are sufficient to support the manuscript conclusions. Therefore, pending the moderate writing style editing, I would recommend the manuscript to be accepted.
This manuscript has been proof read by a professional English editing service for revision.
Reviewer 3 Report
The study by Shin et al is quite interesting.
Points that need to be addressed:
- L25 . Add "role" after essential
- L31-33. Sentence is doubled
- L144. Please specify units of scale bar
- L147. Please correct "Angiogenetic"
- L207-209. Sentence needs to be rewritten because it is unclear
- L394-398. Please clarify experimental replicate numbers (i.e. number of embryos, of experiments, of culture wells...)
-
Author Response
The study by Shin et al is quite interesting.
Thank you for reviewing this manuscript "Functional Characterization of Endothelial Cells Differentiated from Porcine Epiblast Stem Cells" which submitted to Cells. The manuscript has been revised according to the reviewer’s comments.
Points that need to be addressed:
L25 . Add "role" after essential
We have put “role” in the sentence, as you mentioned.
L31-33. Sentence is doubled
We have removed the repeated twice sentence in the abstract, as you mentioned.
L144. Please specify units of scale bar
We have replaced the correct units of scale bar in L144, as you mentioned.
L147. Please correct "Angiogenetic"
Angiogenetic has been altered into angiogenic, as you mentioned.
L207-209. Sentence needs to be rewritten because it is unclear
The sentence has been changed into ‘although primary ECs have several limitations, such as restricted scalability and high probability of karyotypic defects, ECs have been used in various disease models to explore vascular dysfunction’ in L207-209, as you mentioned.
L394-398. Please clarify experimental replicate numbers (i.e. number of embryos, of experiments, of culture wells...)
The sentence has been changed into ‘relative mRNA levels of OCT-3/4, NANOG, SOX2, quantification of Ki-67 positive cells in culture of differentiation media, sprouts length, branch points and quantification of Dil-Ac LDL uptake assay were analyzed in triplicate and data were presented as means ± SEM’ in L394-398, as you mentioned.
Reviewer 4 Report
Abstract should be improved. Introduction is clear. Results are concisely presented. Images are of average quality. Text editing is necessary (some words are missing, a few phrases are repeated, punctuation is to be checked, in the introduction PSC meaning should be indicated before using the abbreviation).
It is evident that having mature endothelial cells makes it possible to perform functional angiogenesis tests on these cells and to verify the impact that certain pathogens may have on the development of cardiovascular diseases. However, I struggle to understand how mature endothelial cells derived from pig epiblast can be useful for regenerative medicine. In this context, what is the advantage of using this approach over other methods in which mature endothelial cells are more easily obtainable ? Please clarify.
Figure 3B. Please recalibrate the scale bar of the boxed sorted-ECs (it is too long).
Figure 5. Magnification of the images is not the same; it seems that the images of the unsorted ECs were taken at a lower magnification. Please check. Please also clarify the labels outside the images; in the provided complex image, it not clear what are the conditions that have been investigated.
Author Response
Abstract should be improved. Introduction is clear. Results are concisely presented. Images are of average quality. Text editing is necessary (some words are missing, a few phrases are repeated, punctuation is to be checked, in the introduction PSC meaning should be indicated before using the abbreviation).
Thank you for reviewing this manuscript "Functional Characterization of Endothelial Cells Differentiated from Porcine Epiblast Stem Cells" which submitted to Cells. The manuscript has been revised according to the reviewer’s comments. This manuscript has been proof read by a professional English editing service for revision.
It is evident that having mature endothelial cells makes it possible to perform functional angiogenesis tests on these cells and to verify the impact that certain pathogens may have on the development of cardiovascular diseases. However, I struggle to understand how mature endothelial cells derived from pig epiblast can be useful for regenerative medicine. In this context, what is the advantage of using this approach over other methods in which mature endothelial cells are more easily obtainable ? Please clarify.
In our experiment using immunodeficiency mice, many results from the in vivo experiment are quite similar with in vitro results. I think that sorted endothelial cells derived from porcine epiblast stem cells have been shown functional characterization of mature endothelial cells in in vivo and in vitro. However, induced pluripotent stem cells have been not differentiated into endothelial cells due to consistent expression of inserted pluripotency genes.
Figure 3B. Please recalibrate the scale bar of the boxed sorted-ECs (it is too long).
The scale bars of the boxed sorted-ECs in Fig 3B have been recalibrated, as you mentioned.
Figure 5. Magnification of the images is not the same; it seems that the images of the unsorted ECs were taken at a lower magnification. Please check. Please also clarify the labels outside the images; in the provided complex image, it not clear what are the conditions that have been investigated.
Magnification of the images and the labels outside images in Fig 5 have been replaced into at a same magnification and correct labels, as you mentioned.
Round 2
Reviewer 1 Report
same as before